# High Meat Consumption Is Prospectively Associated with the Risk of Non-Alcoholic Fatty Liver Disease and Presumed Significant Fibrosis

**DOI:** 10.3390/nu14173533

**Published:** 2022-08-27

**Authors:** Dana Ivancovsky-Wajcman, Naomi Fliss-Isakov, Laura Sol Grinshpan, Federico Salomone, Jeffrey V. Lazarus, Muriel Webb, Oren Shibolet, Revital Kariv, Shira Zelber-Sagi

**Affiliations:** 1School of Public Health, University of Haifa, Haifa 3498838, Israel; 2Department of Gastroenterology Tel-Aviv Medical Center, Tel Aviv 6423906, Israel; 3Division of Gastroenterology, Ospedale di Acireale, Azienda Sanitaria Provinciale di Catania, 95123 Catania, Italy; 4Barcelona Institute for Global Health (ISGlobal) Hospital Clinic, University of Barcelona, 08003 Barcelona, Spain; 5Sackler Faculty of Medicine, Tel Aviv University, Tel Aviv 6997801, Israel

**Keywords:** diet, fatty liver, red meat, processed meat, liver fibrosis, NAFLD

## Abstract

Non-alcoholic fatty liver disease (NAFLD) has been associated with meat consumption in cross-sectional studies. However, only a few prospective studies have been conducted, and they did not test for liver fibrosis. We aimed to assess the association between meat consumption changes and the incidence and remission of NAFLD and significant liver fibrosis. We used a prospective cohort study design, including 316 subjects aged 40–70 years, participating in baseline and follow-up evaluations at Tel-Aviv Medical Center. NAFLD was determined by liver ultrasound or controlled attenuation parameter (CAP), and liver fibrosis was determined by FibroScan. Meat consumption (g/day) was assessed by a food frequency questionnaire (FFQ). In multivariable-adjusted analyses, high consumption of red and/or processed meat (≥gender-specific median) was associated with a higher risk of NAFLD with elevated alanine aminotransferase (ALT) (OR = 3.75, 1.21–11.62, *p* = 0.022). Consistently high (in both baseline and follow-up evaluations) total meat consumption was associated with 2.55-fold (95% CI 1.27–5.12, *p* = 0.009) greater odds for new onset and/or persistence of NAFLD compared to consistently low meat consumption. A similar association was shown for consistently high consumption of red and/or processed meat (OR = 2.12, 95% CI 1.11–4.05, *p* = 0.022). Consistently high red and/or processed meat consumption was associated with 4.77-fold (95% CI 1.36–16.69, *p* = 0.014) greater odds for significant fibrosis compared to consistently low consumption. Minimizing the consumption of red and/or processed meat may help prevent NAFLD and significant fibrosis.

## 1. Introduction

The non-alcoholic fatty liver disease (NAFLD) spectrum includes hepatic steatosis, non-alcoholic steatohepatitis (NASH), fibrosis, and cirrhosis. NAFLD is the most common form of liver disease, with an estimated global prevalence of 25.2% [1]. The fibrosis stage is the strongest predictor of liver-related outcomes and mortality in patients with NAFLD [2,3], and a patient’s knowledge of the fibrosis stage improves adherence to lifestyle changes [4]. Dietary patterns are associated with NAFLD [5,6], and specific foods or dietary components have been identified as promoting or counteracting NAFLD’s progression to fibrosis [7,8,9,10]. In recent years, the association of meat consumption with various adverse health outcomes has been increasingly studied. In cross-sectional and case–control studies, high red meat consumption has been associated with NAFLD [10,11] or NAFLD-related cirrhosis [12]. In addition, two cross-sectional studies demonstrated an association between red meat intake and the degree of liver fibrosis as measured non-invasively by FibroScan, or by liver biopsy [13,14]. However, both studies had no separate analysis for processed meat. A few prospective cohorts among diverse populations found an association between intake of red meat—separated into unprocessed and processed—and steatosis [15,16], but the association with fibrosis was not studied. Moreover, there is a lack of prospective data about the association between changes in meat consumption and the incidence or remission of NAFLD. Therefore, the aim of this study was to assess the association of consumption and changes in consumption of different meat types with the incidence, persistence, and regression of NAFLD, and to evaluate the association of meat intake with liver fibrosis as assessed non-invasively by FibroScan.

## 2. Materials and Methods

### 2.1. Study Design and Population

A prospective cohort study was conducted, including subjects who participated in a baseline metabolic and hepatic screening survey and were followed for at least five years. The baseline survey was conducted between the years 2010 and 2015, and the follow-up evaluation was carried out between the years 2017 and 2020. Exclusion criteria at both time points included the presence of hepatitis B surface antigen (HBsAg) or hepatitis C virus (HCV) antibodies, fatty liver suspected to be secondary to hepatotoxic drugs, inflammatory bowel disease, celiac disease, and/or excessive alcohol consumption (≥30 g/day (d) in men or ≥20 g/day in women) [17,18]. Subjects who reported an unreasonable caloric intake below or above the accepted range (in one or both of the evaluations) 800–4000 Kcal/day for men and 500–3500 Kcal/day for women—were also excluded [19] (Figure 1). The Tel-Aviv Medical Center’s institutional review board (IRB) committee approved the study, and all patients signed informed consent forms.

### 2.2. Data Collection and Definitions of Variables

In both baseline and follow-up evaluations, participants underwent fasting blood tests, liver ultrasound (US) and/or FibroScan, and a face-to-face interview based on a structured questionnaire assembled by the Israeli Ministry of Health for national surveys [20], including demographic details, health status, alcohol and coffee consumption, smoking, and physical activity habits. In addition, they completed a food frequency questionnaire (FFQ), including detailed questions regarding meat consumption [21]. The participants were informed of their US or FibroScan and blood test results only after completing the questionnaires, so as to avoid reporting bias.

### 2.3. NAFLD and Liver Fibrosis Evaluation

Fatty liver was evaluated at baseline by liver US in all patients, while at follow-up the liver US was available only to a subsample (due to availability limitations). FibroScan was available only at the follow-up evaluation. At the follow-up evaluation, one of two methods was used to detect NAFLD: (1) liver US using standardized uniform criteria [22], performed by the same operator and using the same equipment (EUB-8500 scanner Hitachi Medical Corporation, Tokyo, Japan) at both baseline and follow-up; or (2) controlled attenuation parameter (CAP) performed by the same operator and using the same equipment (FibroScan 502 Touch; Echosens, Paris, France), with a cutoff of ≥294 dB/m indicating fatty liver [23]. Persistent NAFLD was defined as a diagnosis at both time points by either modality.

Non-invasive assessment of liver fibrosis was performed by liver stiffness measurements (LSMs) using vibration-controlled transient elastography (VCTE), which has good diagnostic accuracy in the evaluation of fibrosis [24]. The median of 10 measurements represented the LSM score. This was considered reliable only if at least 10 successful acquisitions were obtained and the interquartile range (IQR)-to-median ratio was ≤0.3 [25]. Significant fibrosis was regarded as a value ≥ 8.2 Kpa [23].

NAFLD patients with elevated alanine aminotransferase (ALT), which suggests hepatocellular injury [26], may have a higher inflammatory component [27,28,29,30]. Therefore, we added to the outcomes a subgroup of NAFLD patients who also manifested elevated ALT. Elevated ALT was defined according to the American College of Gastroenterology (ACG)’s clinical guideline cutoffs: ALT > 33 IU/l for men and ALT > 25 IU/l for women [26]. Persistent NAFLD with elevated ALT was defined as NAFLD diagnosis on imaging plus elevated ALT at both time points.

### 2.4. Evaluation and Definitions of Nutritional Variables

The semi-quantitative FFQ—assembled by the Food and Nutrition Administration of the Ministry of Health, and tailored to the Israeli population—was composed of 117/183 (for baseline and follow-up evaluations, respectively; the differences stem mostly from detailed meat preparation methods in the follow-up evaluation) food items, including different meat types with specified serving sizes. We calculated meat consumption in grams (g) per day for each subject at the two time points. Meat types were categorized as previously described [11]; a detailed list of meat variables is depicted in the Appendix (Table A1). High meat consumption was considered above the baseline and follow-up gender-specific medians, as detailed in Table A1. Changes in meat consumption were calculated through four categories: consumption below the gender-specific medians in both time points (consistently low), consumption above the gender-specific median at baseline and below the gender-specific median at follow-up (decreased), consumption below the gender-specific median at the baseline and above the gender-specific median at follow-up (increased), and consumption above the gender-specific median at both time points (consistently high).

### 2.5. Statistical Analysis

Statistical analyses were performed using SPSS version 27 (IBM-SPSS Armonk, NY, USA). Continuous variables are presented as means ± SD. The independent samples *t*-test was used to test differences in continuous variables between the two groups of high and low meat consumption. Associations between nominal variables were tested by Pearson’s chi-squared test, and *p* for trend was calculated when appropriate. A multivariable logistic regression analysis was performed to test the adjusted association between meat intake and the incidence, persistence, and remission of NAFLD, adjusting for potential confounders (i.e., variables that are related to NAFLD, and which differed between the meat intake categories at baseline). For the outcomes “incidence of NAFLD” or “incidence of NAFLD with elevated ALT”, only subjects without these outcomes at the baseline survey were included. For the combined outcome of either “new onset or persistence” (the presence of outcome at both time points) of “NAFLD” or “NAFLD with elevated ALT”, the entire sample was included in the analysis. In this analysis, the comparison was made to subjects who had never had these outcomes, or had a remission of the outcome at the follow-up evaluation. The fully adjusted model includes both potential confounders (i.e., age, gender, energy, body mass index (BMI)) and potential mediators (i.e., protein and cholesterol intake). The odds ratio (OR) and 95% confidence interval (CI) are presented. A *p*-value ≤ 0.05 was considered statistically significant for all analyses.

## 3. Results

### 3.1. Description of the Study Population and Comparison between Subjects with High and Low (by Gender-Specific Median) Meat Consumption

A total of 970 subjects participated in the baseline survey. Of those, 402 attended the follow-up evaluation. Nineteen subjects were excluded because of hepatotoxic drug use, secondary liver diseases, or other related medical conditions. Sixty-three subjects were excluded because of unreasonable caloric intake at either baseline or follow-up evaluation. Of the 320 subjects remaining, only 316 were assessed for NAFLD in the follow-up evaluation, and were included for analysis (101 subjects underwent the liver US, 236 underwent CAP) (Figure 1). The mean time to follow-up was 6.79 ± 0.67 (range of 5.21–8.47) years.

In the final sample, 179 subjects were male (56.60%), the mean age at baseline was 58.65 ± 6.44 years, and the mean BMI was 28.12 ± 5.48 Kg/m^2^.

New onset or persistence of NAFLD was found in 34.50% (*n* = 109/316) of the sample. Of those without NAFLD at baseline, 18.20% (*n* = 36/198) had a new-onset NAFLD. Remission of NAFLD occurred among 38.10% (*n* = 45/118) of those with an NAFLD diagnosis at baseline. Presumed significant fibrosis evaluated by LSM was detected in 10.20% of the sample in the follow-up evaluation (*n* = 24/236).

At baseline, subjects with high meat consumption had a worse metabolic profile, including higher serum glucose levels and homeostasis model assessment for insulin resistance (HOMA-IR). Subjects with high meat consumption also had higher caloric intake, cholesterol, and protein as a percentage of total calories (Table 1).

### 3.2. Multivariable Association of High Meat Consumption with NAFLD, and of NAFLD with Elevated ALT

There was no significant association between meat consumption of any type and NAFLD. However, high consumption of red and/or processed meat was associated with a new onset/persistence (OR = 3.07, 95% CI 1.31–7.21, *p* = 0.010) or incidence of NAFLD with elevated ALT (OR = 3.75, 95% CI 1.21–11.62, *p* = 0.022), adjusting for the following potential confounders and mediators: baseline age (years), gender, BMI (Kg/m^2^), energy (Kcal), protein (% total Kcal), and cholesterol intake (mg/day). High processed meat consumption was associated with new onset/persistence of NAFLD with elevated ALT (OR = 2.52, 95% CI 1.14–5.59, *p* = 0.023). Likewise, unprocessed red meat consumption was positively associated with new onset/persistence of NAFLD with elevated ALT (OR = 2.28, 95% CI 1.04–4.99, *p* = 0.039) (Table 2). We did not find associations between meat consumption and remission of any outcomes (data not shown).

### 3.3. Univariate and Multivariable Association between Changes in Consumption of Different Meat Types and NAFLD

In a univariate analysis, subjects with high total meat or red and/or processed meat consumption at both the baseline and follow-up evaluations had the highest prevalence of NAFLD compared to those with low meat consumption at one or both evaluations, with a modest dose–response trend across categories (*p* for trend = 0.002 for total meat, and 0.013 for red and/or processed meat) (Figure 2A). A similar trend was also shown in a multivariable analysis adjusting for all potential confounders and mediators. Consistently high total meat consumption was associated with 2.55-fold (95% CI 1.27–5.12, *p* = 0.009) greater odds for new onset/persistence of NAFLD compared to consistently low meat consumption. The same association was shown for the consumption of red and/or processed meat (OR = 2.12, 95% CI 1.11–4.05, *p* = 0.022) (Figure 2B).

### 3.4. Sensitivity Analysis for the Association between Meat Consumption and NAFLD Evaluated Only by Liver US at Both Time Points

A sensitivity analysis found similar associations in a subsample of 101 subjects undergoing liver US at both time points. In a multivariable model, high baseline consumption of total meat and of the different types of meat was associated with new onset/persistence of NAFLD (OR = 4.82, 95% CI 1.48–15.73, *p* = 0.009; OR = 4.51, 1.44–14.11, *p* = 0.010; OR = 3.15, 1.09–9.10, *p* = 0.035; OR = 3.99, 1.31–12.22, *p* = 0.015 for total, red and/or processed, processed, and unprocessed red meat consumption, respectively). In addition, high consumption of unprocessed red meat was significantly associated with lower odds of NAFLD remission (OR = 0.16, 95% CI 0.03–0.79, *p* = 0.024) (Figure A1).

As for changes in meat consumption, an increase of 10% (vs. no change or decrease) in total meat (OR = 3.05, 95% CI 1.09–8.49, *p* = 0.033), red and/or processed meat (OR = 3.23, 1.11–9.42, *p* = 0.032), or unprocessed red meat (OR = 5.41, 1.74–16.78, *p* = 0.003) was associated with new onset/persistence of NAFLD, along with lower odds of NAFLD remission (OR = 0.09, 95% CI 0.01–0.58, *p* = 0.003; OR = 0.21, 0.05–0.93, *p* = 0.039; OR = 0.11, 0.02–0.58, *p* = 0.009, for total meat, red and/or processed meat, and unprocessed meat, respectively). (Figure A1).

### 3.5. Multivariable Association of High Red and/or Processed Meat Consumption at Baseline with Presumed Liver Fibrosis at Follow-Up

In a univariate analysis, the prevalence of presumed significant fibrosis at the follow-up evaluation was the highest among those with consistently high red and/or processed meat consumption, but it was not statistically significant (Figure 3A). However, in a multivariable analysis, consistently high red and/or processed meat consumption was associated with 4.77-fold (95% CI 1.36–16.69, *p* = 0.014) greater odds of presumed significant fibrosis as compared to consistently low meat consumption. There was no association with total meat consumption (Figure 3B).

## 4. Discussion

The harmful effect of high consumption of meat on human health is well established, including the induction of metabolic alterations such as insulin resistance (IR), as well as related diseases such as type 2 diabetes [31], metabolic syndrome [32], cardiovascular diseases [33], and some cancers [34]—especially colorectal cancer [35]. The present study shows a prospective association of total, red, and/or processed meat intake with the incidence and persistence of NAFLD and presumed clinically significant fibrosis. Prior to the results of this prospective study, some cross-sectional studies suggested the adverse effects of animal proteins [36] and a Western diet—which is rich in meat [13]—on the liver. Specifically, among a multiethnic population, a nested case–control study found that higher intakes of red meat, processed red meat, and poultry were associated with an increased risk of NAFLD and/or NAFLD-related cirrhosis [12]. Conversely, in our study, there was no association between poultry or white meat (i.e., poultry and fish) and NAFLD (data not shown). Moreover, among a general population in the United States, a prospective cohort study with a 16-year follow-up showed that high intakes of total meat, processed and unprocessed red meat (i.e., beef, lamb, and pork), and nitrites from processed meat were all independently associated with liver-disease-related mortality. In contrast, total white meat was correlated with reduced all-cause mortality [37]. In addition, a meta-analysis of seven cross-sectional studies (*n* = 5141 cases) demonstrated a positive association between red meat consumption and the risk of NAFLD [10]. Notably, these consistent findings in diverse populations strengthen the external validity of our results.

In our prospective cohort, we also found that high red and/or processed meat consumption was associated with a diagnosis of NAFLD with elevated ALT, suggesting greater hepatocellular injury [26] and an increased likelihood of steatohepatitis [26] among consistent meat eaters. A prospective cohort in the Iranian population found a similar association between high red meat consumption (highest vs. lowest quartiles) and new-onset NAFLD with elevated ALT adjusted for age and gender, but this finding was attenuated after adjustment for other lifestyle habits [15]. A recent cohort study of 77,795 women in the Nurses’ Health Study II cohort found a dose–response association between red meat intake and the risk of developing NAFLD. However, after further adjustment for BMI, all associations were attenuated [16], pointing to the confounding effect of BMI. However, in our study, the association was independent of BMI, suggesting that obesity is not the primary determinant of our findings.

Furthermore, as assessed by liver stiffness, we demonstrated an association between red and/or processed meat consumption and clinically significant fibrosis. Our results are consistent with those of previous cross-sectional and case–control studies. In a cross-sectional study of 170 subjects with NAFLD, high red meat consumption was associated with a twofold increased risk of greater liver stiffness, after adjustment for lifestyle habits [13]. In another cross-sectional study of 94 morbidly obese subjects with NAFLD, high red meat and low carbohydrate consumption were independently associated with fibrosis evaluated by liver biopsy [14], but without adjustment for body fat measures or other lifestyle risk factors.

Several molecular determinants might explain the association between meat intake and NAFLD and fibrosis, such as high contents of saturated fats and cholesterol, heme iron, and unfavorable meat products created after specific cooking methods such as advanced glycation end products (AGEs), heterocyclic amines, and other muscle protein oxidation byproducts [11,13,38,39]. Oxidation of saturated fats and cholesterol induces the production of reactive dialdehydes and isoprostanes, which promote inflammation and fibrogenesis in the liver [40,41,42]. Similarly, AGEs [43] and excess iron [44] are also triggers of hepatic stellate cell activation. In addition, the gut microbiome may also contribute to the association found in our study. In fact, high meat consumption may promote microbiota dysbiosis [45], which is implicated in the onset and progression of NAFLD [46]. Finally, preservative contents, nitrites, or nitrates used in meat processing were previously demonstrated to be associated with NAFLD [37] and liver fibrosis [47], although without fully clear mechanisms.

It may be claimed that heavy meat eaters tend to eat a generally less healthy diet or have a more unhealthy lifestyle. However, when we adjusted for nutritional and lifestyle parameters, which were found to be different between frequent and infrequent meat eaters, it did not attenuate the associations, indicating that these factors do not entirely explain the association of meat intake with NAFLD. We did find that heavy meat eaters have higher HOMA-IR—a marker of IR, which is a well-established key factor in NAFLD’s pathogenesis [48], and may explain some of the observed associations.

The strengths of our study include its prospective study design with two repeated comprehensive liver evaluations enabling us to examine the incidence, remission, and persistence of NAFLD. We also applied detailed meticulous assessments of meat intake at both time points, enabling the evaluation of changes in meat intake during follow-up. Finally, we evaluated liver fibrosis in addition to the evaluation of liver fat. However, this study has several limitations that should be discussed. First, dietary habits were self-reported, which may have led to a reporting bias. Nevertheless, since the participants and the research team were blinded to the liver and blood test results at baseline and due to the prospective nature of the study, this is a non-differential bias, and may have only led to underestimation of the observed associations. Second, the diagnosis of NAFLD was determined by liver US or CAP versus liver histology, which is impossible to obtain in a study among the general population. In addition, the diagnostic method of NAFLD was changed in the follow-up study, and although all methods were well-validated, this could create an information bias. Therefore, we performed a sensitivity analysis of the subsample of the subjects who underwent an US examination at the two time points, and the associations were similar to those observed in the entire sample. Third, liver fibrosis was evaluated by FibroScan and not by liver biopsy, which remains the reference standard for histological evaluation of NASH and fibrosis, but its use is limited due to its invasive nature and sampling error. We used this approach instead of a highly validated non-invasive alternative of liver stiffness, which has good diagnostic accuracy for the assessment of liver fibrosis [24,49,50].

Our results add to the evidence on the importance of nutrition in the prevention and treatment of NAFLD. They also support the current recommendation to limit red and processed meat intake for the good of general health. The 2019 American guidelines on the primary prevention of cardiovascular disease recommend minimizing the intake of red meat and processed red meats [51]. The dietary recommendations for cardiometabolic health provide a more quantitative limit of intake of processed meats and unprocessed red meats—approximately one serving/w of 50 g and no more than 1–2 servings/w of 100 g, respectively [52]. This study also emphasizes the necessity of a multidisciplinary team, including dietitians, responsible for comprehensive management of patients with NAFLD [53].

## 5. Conclusions

Red and processed meat consumption and changes in consumption over time are associated with NAFLD and liver fibrosis; therefore, people with fatty liver disease should be recommended to minimize their consumption of these meats in addition to other interventions.

## Figures and Tables

**Figure 1 nutrients-14-03533-f001:**
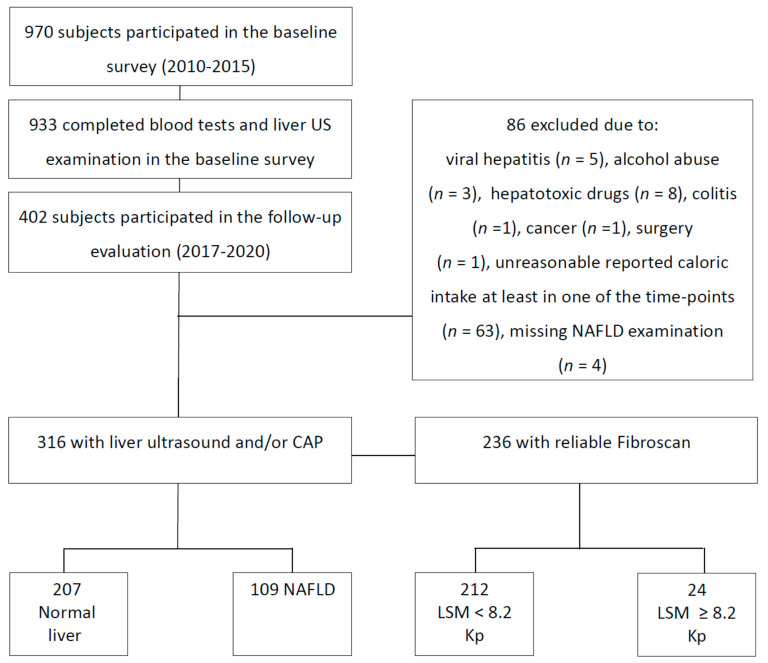
Flowchart of the study population. Abbreviations: US, ultrasound; NAFLD, non-alcoholic fatty liver disease; CAP, controlled attenuation parameter, LSM, liver stiffness measurement.

**Figure 2 nutrients-14-03533-f002:**
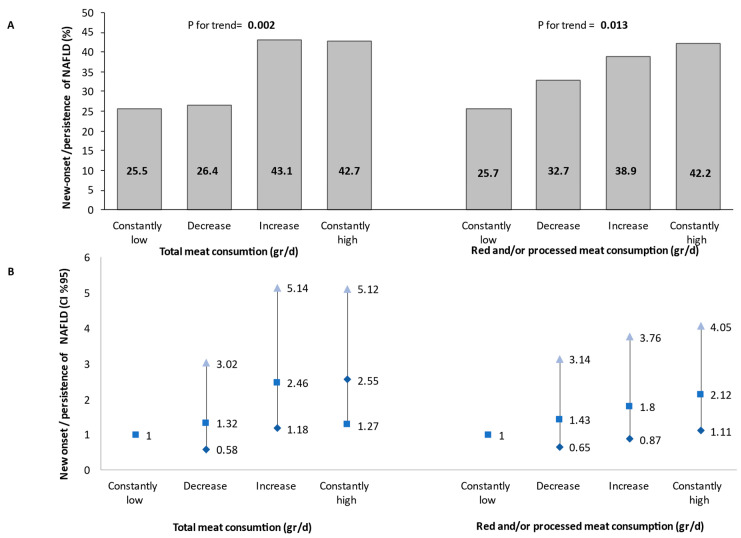
Univariate (**A**) and multivariable (**B**) association between changes in meat consumption during follow-up and new-onset or persistent NAFLD (as determined by either liver US or CAP). The multivariable analysis model was adjusted for baseline age (years), gender, BMI (Kg/m^2^), energy (Kcal), protein (% total Kcal), and cholesterol intake (mg/day). *Consistent low consumption*: consumption below the gender-specific medians at both the baseline and follow-up evaluations (N total in the category = 102/105 for total or red and/or processed meat, respectively). *Decreased*: consumption above the gender-specific median at the baseline survey and below the gender-specific median at the follow-up evaluation (N total in the category = 53/55 for total or red and/or processed meat, respectively). *Increased*: consumption below the gender-specific median at the baseline survey and above the gender-specific median at the follow-up evaluation (N total in the category = 58/54 for total or red and/or processed meat, respectively). *Consistent high consumption*: consumption above the gender-specific median at both the baseline and follow-up evaluations (N total in the category = 103/102 for total or red and/or processed meat, respectively). Abbreviations: NAFLD, non-alcoholic fatty liver disease; CI, confidence interval; d, day.

**Figure 3 nutrients-14-03533-f003:**
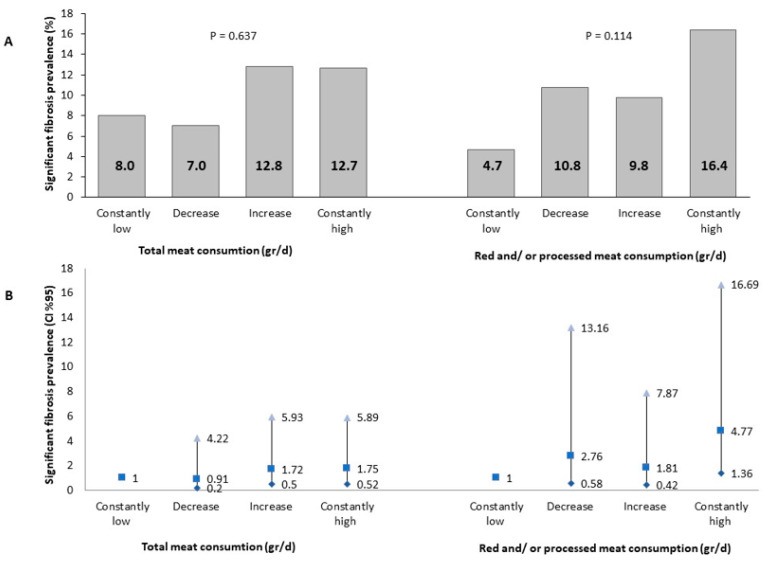
Univariate (**A**) and multivariable (**B**) association between meat consumption and significant fibrosis (LSM ≥ 8.2 Kp). Abbreviations: CI, confidence interval. The multivariable analysis model is adjusted for baseline age (years), gender, BMI (Kg/m^2^), energy (Kcal), protein (% total Kcal), and cholesterol intake (mg/day). *Consistent low consumption*: consumption below the gender-specific medians at both the baseline and follow-up evaluations (N total in the category = 75/85 for total or red and/or processed meat, respectively). *Decreased:* consumption above the gender-specific median at the baseline survey and below the gender-specific median at the follow-up evaluation (N total in the category = 43/37 for total or red and/or processed meat, respectively). *Increased:* consumption below the gender-specific median at the baseline survey and above the gender-specific median at the follow-up evaluation (N total in the category = 47/41 for total or red and/or processed meat, respectively). *Consistent high consumption:* consumption above the gender-specific median at both the baseline and follow-up evaluations (N total in the category = 71/73 for total or red and/or processed meat, respectively). Abbreviations: CI, confidence interval; d, day.

**Table 1 nutrients-14-03533-t001:** Baseline characteristics of participants according to meat consumption.

Variable	Baseline Gender-Specific Medians of Total Meat Consumption		
	Low Consumption ^1^ (*n* = 160)	High Consumption ^1^ (*n* = 156)	*p*-Value
Age (years)	59.06 ± 6.25	58.22 ± 6.63	0.244
Gender (% male)	56.90	56.40	0.934
BMI (kg/m^2^)	27.79 ± 5.77	28.45 ± 5.17	0.288
Weight change % ^2^	−0.18 ± 12.04	−1.22 ± 7.26	0.355
Glucose (mg/dl)	87.04 ± 17.03	92.48 ± 21.36	0.013
HbA1C (%) (*n* = 309)	5.76 ± 0.61	5.84 ± 0.74	0.285
HOMA-IR (score)	2.50 ± 1.63	3.00 ± 2.24	0.024
Total cholesterol (mg/dl)	182.72 ± 37.36	180.22 ± 32.17	0.526
Triglycerides (mg/dl)	110.62 ± 65.18	114.28 ± 56.73	0.596
ALT (U/L)	25.81 ± 10.67	28.40 ± 20.33	0.160
AST (U/L)	26.39 ± 8.02	24.61 ± 9.13	0.068
GGT (U/L)	25.37 ± 17.76	31.20 ± 32.53	0.052
Uric acid (mg/dl)	5.44 ± 1.38	5.53 ± 1.41	0.569
Ferritin (ng/mL) (*n* = 293)	85.34 ± 68.95	94.24 ± 89.02	0.338
Dietary intake and lifestyle habits			
Energy (Kcal)	1869. 95 ± 681.17	2134.61 ± 631.44	<0.001
Protein (% total Kcal)	17.26 ± 4.29	20.69 ± 4.65	<0.001
Carbohydrates (%total Kcal)	43.08 ± 8.79	39.12 ± 8.30	<0.001
Fat (% total Kcal)	35.93 ± 6.32	36.85 ± 6.16	0.189
Saturated fatty acids (% total Kcal)	12.56 ± 3.73	12.48 ± 3.33	0.835
Cholesterol (mg/day)	270.10 ± 130.44	407.12 ± 215.84	<0.001
Coffee (cup/day)	3.00 ± 2.96	2.78 ± 2.76	0.514
Fiber (g/day)	23.89 ± 13.04	22.88 ± 9.69	0.437
Sugared beverages (cups/day)	1.82 ± 3.01	1.88 ± 3.53	0.865
Smoking (% ever smokers)	50.60	49.40	0.822
Physical activity (h/week)	2.73 ± 3.49	2.43 ± 2.95	0.404
Alcohol (portions/week)	1.69 ± 2.57	1.95 ± 3.00	0.414

Abbreviations: BMI, body mass index; HOMA-IR, homeostasis model assessment for insulin resistance; ALT, alanine aminotransferase; AST, aspartate aminotransferase; GGT, gamma glutamyl transferase. ^1^ High meat consumption defined above the gender-specific medians: total meat cutoff ≥ 88.2 g/day among women and 122.9 g/day among men. ^2^ Weight change calculated as the percentage of change (in Kg) from baseline: (weight in follow up minus weight in baseline)/weight in baseline × 100.

**Table 2 nutrients-14-03533-t002:** Multivariable analysis of the association between high meat consumption (above gender-specific medians) at baseline and incidence or persistence of NAFLD at follow-up.

	New Onset or Persistence (vs. Never or Remission)	Incidence (New Onset among Those without the Outcome at Baseline)
	OR (95% CI), *p*-Value	
**NAFLD**		
	N cases/N total (109/316)	N cases/N total (36/198)
Total meat (≥88.2 g/day women/≥122.9 men)	1.41 (0.81–2.46), 0.230	1.37 (0.58–3.23), 0.472
Red and/or processed meat (≥16.3 g/day women/≥37.2 men)	1.51 (0.89–2.56), 0.129	1.28 (0.56–2.96), 0.557
Processed meat (≥1.8 g/day women/≥5.7 men)	1.17 (0.71–1.93), 0.545	1.12 (0.53–2.40), 0.767
Unprocessed red meat (≥9.6 g/day women/≥26.2 men)	1.41 (0.85–2.34), 0.181	0.93 (0.41–2.06), 0.848
**NAFLD with elevated ALT ^1^**		
	N cases/N total (34/314 ^2^)	N cases/N total (19/275)
Total meat (≥88.2 g/day women/≥122.9 men)	1.18 (0.52–2.69), 0.694	1.77 (0.60–5.26), 0.301
Red and/or processed meat (≥16.3 g/day women/≥37.2 men)	3.07 (1.31–7.21), 0.010	3.75 (1.21–11.62), 0.022
Processed meat (≥1.8 g/day women/≥5.7 men)	2.52 (1.14–5.59), 0.023	2.22 (0.80–6.14), 0.124
Unprocessed red meat (≥9.6 g/day women/≥26.2 men)	2.28 (1.04–4.99), 0.039	2.62 (0.93–7.36), 0.068

Abbreviations: NAFLD, non-alcoholic fatty liver disease; OR, odds ratio; CI, confidence interval; ALT, alanine aminotransferase; BMI, body mass index; AGC, American College of Gastroenterology. All models are adjusted for baseline age (years), gender, BMI (Kg/m^2^), energy (Kcal), protein (% total Kcal), and cholesterol intake (mg/day). ^1^ Elevated ALT was defined as ALT > 33 IU/l for men and ALT > 25 IU/l for women according to the ACG clinical guidelines. ^2^ Only 314 subjects had ALT measurements.

## Data Availability

The data presented in this study are available upon request from the corresponding author. The data are not publicly available due to privacy and ethical restrictions.

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
