# Peer review of "High Meat Consumption Is Prospectively Associated with the Risk of Non-Alcoholic Fatty Liver Disease and Presumed Significant Fibrosis"

_nutrients, 2022, doi:10.3390/nu14173533_

Round 1

Reviewer 1 Report

  1.  The authors should check interaction of associations sex because the amount of meat intake was quite different between sexes.
  2. The authors made too many variables for both dependent variables and independent variables. The variables are inconsistent through the manuscript. They should perform analyses with limited variables, which makes the manuscript easier to read. 

Reviewer 2 Report

Dear authors this is a very interesting study. The manuscript is informative and well written. The methods are clearly presented, as well as the results. The discussion provides the necessary critical appraisal of the results of your study comparing them with the results of other relevant studies. Finally, strengths and limitations are adequately addressed. The only point I would like to comment is on the use of some abbreviations within the text without mentioning before the full term.

Reviewer 3 Report

The purpose of this manuscript is to evaluate the association between the consumption and variations in the consumption of different types of meat with the incidence, persistence and regression of NAFLD and to evaluate the association between meat intake and liver fibrosis assessed non-invasively using Fibroscan.

The study was conducted well but there are some issues to be resolved before possible publications. 

1. The design of the paper is not very clear. The authors write that they used two "prospective cohorts" but the data date back crica 10 years. The authors must clarify whether this study is retrospective or prospective.

2. There are profound differences in caloric intake in the two groups divided by how much meat they eat. The authors correctly ajusted the data trying to take this difference into account. However, it should be clarified why there is this difference and how much the different BMi between the two groups affects the results. 

3. More emphasis should be placed in the discussion on the correlated risk between white, red and processed meat. 

4. It is not among the main otcome of the paper, but since the authors have the data, is it possible that there is a correlation in a protective sense between the consumption of plant-origin prteins and fibre and the risk of NAFLD and liver fibrosis?

5. There are some plagiarised paragraphs, see attachment. The number of self-citations is exaggerated. 

Round 2

Reviewer 1 Report

The reviewer felt the manuscript was well revised.

Interaction between sexes was adequately checked. I recommend the authors to mention it in Result because most of epidemiologists concern this problem.

I agree with the withdraw of Table 3 in the previous manuscript which included a problem that FFQ is not good for quantity evaluation. The revised manuscript is also easier to read.

Reviewer 3 Report

The authors fixed the paper and answered all my comments. 

Author Response

Thank you for your comment